# Psychosocial Impacts of the COVID-19 Pandemic on Women with Spinal Cord Injury

**DOI:** 10.3390/ijerph20146387

**Published:** 2023-07-18

**Authors:** Heather B. Taylor, Rosemary B. Hughes, Diana Gonzalez, Muna Bhattarai, Susan Robinson-Whelen

**Affiliations:** 1TIRR Memorial Hermann, Houston, TX 77030, USA; 2Department of Physical Medicine and Rehabilitation, University of Texas Health Science Center, Houston, TX 77054, USA; 3Rural Institute for Inclusive Communities, University of Montana, Missoula, MT 59812, USA; 4Texas A&M University, College Station, TX 77843, USA; 5Department of Physical Medicine and Rehabilitation, Baylor College of Medicine, Houston, TX 77030, USA

**Keywords:** COVID-19, pandemic, mental health, impacts, women, spinal cord injury

## Abstract

This study represents the first known research addressing the impact of the COVID-19 pandemic on women with spinal cord injury (SCI) in the United States. Women in this population face unique barriers that put them at elevated risk for compromised quality of life, risk that was magnified by physical and social restrictions imposed during the pandemic. This qualitative study examined the perceptions of women with SCI and the effect of the pandemic on their lives. The predominantly White and relatively well-educated sample of 105 women with traumatic SCI was diverse in age, injury characteristics, and geographic representation. Recruited across the USA, participants in an online psychological health intervention trial were asked to respond to the item, “Please tell us how COVID-19 has affected you and your life”, administered May–October, 2020. An overall sentiment rating of impact was coded as well as the impact of COVID-19 on eight individual themes: Physical Health, Mental Health, Social Health, Activities of Daily Living, Exercise, Work, Activities Outside the Home, and Activities at Home. Sentiment responses were rated as positive, negative, a mixture of positive and negative impacts, or neutral impact. Participants described the overall impact of COVID-19 as negative (54%), positive (10%), mixed (21%) or neutral (15%). Sentiment ratings to individual themes were also described. Our findings highlight the importance of providing access to disability-sensitive and affordable support, resources, and interventions for women with SCI, especially during a public health crisis.

## 1. Introduction

The COVID-19 pandemic exerted profound disruptions on the lives of people worldwide, with differential ramifications for women and people with disabilities. Amplifying pre-existing gender disparities, the pandemic heightened levels of gender-based violence, intensified lack of employment, and increased unpaid child and elderly care work among women [1]. A highly marginalized and underserved community, people with disabilities experienced additional and unique impacts [2,3,4,5,6]. Of significance, they faced increased barriers to accessing health care, personal assistance with essential activities of daily life, employment, accessible messaging about the pandemic, and other essential supports and resources [5]. When compared to people without disabilities, they experienced greater negative effects involving financial hardship [7] and psychological distress, including loneliness and a sense of hopelessness [8,9].

A large population experiencing historical inequities and marginalization [10], women with disabilities faced unique pandemic-related oppression and disadvantage. According to the findings of a global assessment of the pandemic’s impact on women with disabilities [11], policies and programs designed to address the pandemic failed to adequately address the needs of women with disabilities, exacerbating their existing health disparities and difficulties in meeting basic needs [11]. With a historically disproportionate prevalence of interpersonal violence, women at the intersection of gender and disability were at even greater risk of violence during the pandemic [11].

In the United States, approximately 37.5 million women, or about one in four women, report having a disability [12]. Disability has been defined in terms of core functional limitations in seeing, hearing, walking, cognition, self-care, and communication [13]. Spinal cord injury (SCI) is a profound, life-changing event often associated with significant functional limitations and disability resulting from the loss of sensory, motor, and autonomic control [14]. People with SCI also risk developing urinary tract infections, pressure ulcers and other potentially life-threatening secondary complications that result largely from inadequate medical care as well as physical, social, and policy barriers to accessing healthcare [14]. Women represent only about one in five (22%) of the approximate 299,000 people living with SCI in the United States [15]. As such, gender differences and gender-specific issues that affect outcomes in women with SCI are often overlooked or ignored by researchers.

The growing literature on the effects of the pandemic on people with SCI strongly suggests that people in this population have experienced significant adverse impacts, including elevated risks for exposure to the virus [16], severe illness, and COVID-19-related mortality [17]. Other negative impacts include concerns about medical rationing, fear of contracting the virus, pandemic-driven health anxiety, emotional distress—including depression and social isolation—as well as substantial barriers to accessing medical supplies, food security, healthcare services, and personal assistance services [18,19,20,21,22,23,24]. Robinson-Whelen and colleagues [25] examined self-reported impacts of the pandemic on 346 adults with SCI (42% women) who responded to items related to COVID-19’s impact on their psychosocial health, physical health and health behaviors, and social participation. The data were collected toward the end of the first year of the pandemic. Although many reported no change due to the pandemic, among those reporting change, more reported negative versus positive change. In fact, nearly half reported increased stress, depression, and loneliness, and less in-person interaction and participation in life roles, with women with SCI reporting greater adverse impacts on their psychosocial health than men with SCI.

Despite the amplitude of negative impacts of the pandemic, some unexpected positive outcomes have been observed among the general population [26,27,28]. For example, the pandemic appears to have prompted a greater focus on self-care, health awareness, and time with family and other relationships [26]. Although “stay at home” orders significantly disrupted in-person social involvement and other day-to-day functioning, research suggests those orders also resulted in people spending more time at home, minimizing work- and commute-related stress, and learning to rely on telehealth, e-learning, social media, and advanced technological systems [27]. Although experiencing multiple disastrous impacts during the pandemic, the disability community has reportedly benefited in at least one area, that is, improved employment during the pandemic recovery period, especially in occupations allowing for telework [29]. Others may have reduced the deleterious impacts of the pandemic by relying on their coping skills. For example, while most participants with SCI in a qualitative study [30] reported profound challenges related to meeting their complex care needs and avoiding the virus, some indicated that dealing with the pandemic may have been facilitated by having learned to cope with the challenges of living with SCI. Similar research indicated that although most participants with physical disabilities described fear, isolation, and invisibility during the pandemic, some reported only minor disruptions to their daily lives [31].

Although some attention has been given to the COVID-19 pandemic-related disparities facing women with disabilities and mixed samples of men and women with SCI, there has been no investigation on the unique impacts of the pandemic on women with SCI. The purpose of this qualitative study was to explore the perceived positive and negative impacts of the COVID-19 pandemic in a sample of women with SCI in the United States during the first year of the pandemic.

## 2. Methods

### 2.1. Procedures and Recruitment

This study consists of analyses of data collected as part of a randomized controlled trial of a group psychological health promotion intervention for women with SCI living in the United States. The study was underway at the time of the onset of the COVID-19 pandemic. Given the potential impact of the COVID-19 pandemic on mental health, we decided to add one open-ended question about the impacts of the pandemic to the existing study survey. The data reported here represent responses to the first survey participants completed after COVID-19 was declared a pandemic. For two cohorts, the data reflect responses from their baseline survey; for one cohort, their post-intervention survey; and for one cohort, their six-month follow-up survey. For all respondents, the responses reflect the first time they were presented with the open-ended question about the pandemic, and for all participants, data were collected between May and September, 2020.

After obtaining approval from our Institutional Review Board, we recruited nationally by distributing study announcements through local and national disability, healthcare, or community service organizations, as well as through our center’s database of women with SCI who had given prior consent to be contacted about research opportunities. We invited interested individuals to contact us by either phone or email. We then conducted a screening interview to determine eligibility and obtain consent. Women were eligible if they were 18 years of age or older; had a traumatic SCI for at least one year; used a mobility device; had access to a computer, email, and the Internet; and resided in the community versus an institution. Exclusion criteria included the inability to provide consent, complete questionnaires, and participate in the intervention in English; self-reported misuse of alcohol or other drugs; suicidal ideation with intent; and sensory impairment that prohibited participation in the intervention. We also required individuals to demonstrate informed consent by correctly answering questions about the study.

### 2.2. Measures

The online survey was created using Qualtrics and was designed to be accessible to all eligible women, including those with limited hand function who utilized voice-activated software. Participants were also offered alternative methods of completing the online survey (e.g., paper and pencil, telephone interview). The survey included demographic and disability questions. A brief introduction statement on COVID-19 helped orient participants to the topic preceding the single open-ended question about the impacts of the pandemic.

#### 2.2.1. Demographic and Disability Characteristics

Demographic characteristics collected included age, race/ethnicity, relationship status, education, employment, household income, and rural vs. urban setting. Disability-related characteristics included level of injury, time since injury, primary method of ambulation, and need for assistance with daily activities.

#### 2.2.2. Assessment of the Impacts of the COVID-19 Pandemic

Women with SCI participating in the study were given the following introduction to the COVID-19 question: “We know that COVID-19 can affect nearly every aspect of people’s lives. Some become ill themselves or their family members become infected with the virus. Some experience isolation associated with social distancing. Many experience losses related to the death of loved ones, financial hardship, or difficulty accessing healthcare and maintaining personal assistance. As a result, people may experience other physical and emotional health problems. We have, however, also heard from those who say they appreciate the time to connect with others electronically or focus on activities or hobbies. Because these COVID-related experiences can affect your responses to some of our survey questions, we thought it was important to have some information about the extent to which COVID-19 has affected you”. This statement was followed by one open-ended question, “Please tell us how COVID-19 has affected you and your life”. Responses to this statement were independently entered by each participant in their own words.

### 2.3. Analysis Plan

#### 2.3.1. Qualitative Analysis: Thematic and Sentiment Analysis

Thematic analysis was used to analyze and organize information and identify recurring patterns across participant responses. This method has been used to draw reasonable and meaningful conclusions from qualitative data and is a widespread method for qualitative analysis [32] (pp. 342–386). This was conducted by reviewing the raw written responses to the COVID-19 question line by line and using a combination of deductive codes to identify themes in participant responses and inductive codes generated by the data. First, participant responses were qualitatively evaluated to determine consistency in the responses. In this way, their responses were grouped into categories or “themes”. Three doctoral-level investigators conducted the thematic coding, with one investigator taking the lead on initial coding and bringing coding issues to the other two investigators. This iterative process continued until agreement was met on identification and definitions of themes.

Next, hand-coded Sentiment Analysis (SA) was conducted. SA is a process of computationally identifying and categorizing opinions expressed in a statement. This is often used to determine whether the individual’s expressed written attitude towards a particular topic is positive, negative, or neutral. For example, SA has been used to evaluate responses to social media information and text during the COVID-19 pandemic in other studies [33,34]. However, no study has evaluated open-ended responses to a question about one’s personal opinion about the impacts of COVID-19. Furthermore, no study has evaluated SA in a sample of women with SCI using hand-coding, which is identified as the most accurate and valid means to gather sentiment information over other methods [35].

#### 2.3.2. Sentiment Coding Training and Procedures

For the purposes of this study, coders manually read through each participant’s statement and assigned ratings indicating if the sentiments written by the participant described a positive, negative, mixed, or neutral (i.e., no change) impact from COVID-19. Two independent coders were trained to conduct sentiment rating. Training continued until coders reached a reliability of at least 80% over five practice trials. Any discrepancies during training were brought back to the investigators for clarification. Coders then began assigning independent ratings of each statement using the following sequence: First, each statement was coded for an Overall Impact sentiment rating. Next, each statement was reviewed again to evaluate the presence of each theme and the specific sentiment attributed to each theme. In this way, coders determined if the theme was present in the statement. Then, if present, coders identified the sentiment in the specific statement(s) associated with the individual theme. Finally, any rating discrepancy between coders was reviewed by a third independent coder to determine the final rating decision. This sequential process assured standardization and reliability among coders.

## 3. Results

### 3.1. Participants

A total sample of 105 women with traumatic SCI responded to the COVID-19 impact question. Three participants of the 108 who received the COVID-19 question left the question blank, choosing not to respond. These participants were not included in the study analyses. The study participants were diverse in terms of age (Mean = 43.20, SD = 13.09, range = 21–75 years of age), time since injury (17.47, SD = 13.06, range = 1–50 years), and level of injury (paraplegia, 53%; tetraplegia, 47%). The sample was predominantly White, non-Hispanic (74%) and relatively well-educated (61% had a college or graduate degree). Two-thirds (67%) reported living in urban areas, and 62% reported being unemployed (See Table 1 for more details).

### 3.2. Themes Identified

Evaluation of responses to the item, “Please tell us how COVID-19 has affected you and your life”, resulted in eight unique themes: (1) physical health, (2) mental health, (3) social health, (4) work, (5) exercise, and (6) activities of daily living (ADL/IADL), participating in (7) activities at home (engaging in activities in the home including inside the house and around the yard) and participating in (8) activities outside of the home (going out and engaging in activities away from the home). For each theme, sentiment specific to the theme was indicated using the four sentiment ratings: (1) positive, (2) negative, (3) mixed, and (4) neutral. If a statement did not discuss one of the eight themes, coders indicated that no theme was present in the statement (e.g., ND—no data). This ensured complete ratings across participants and themes.

### 3.3. Sentiment of Overall Impact

The overall impact of COVID-19 was rated based on the sentiment reflected in the individual’s response in its entirety. All 105 respondents provided codable statements. First, “time of survey completion” was evaluated to determine if there were differences between the sentiment responses of those who were exposed to the intervention, compared to those who were not exposed to the intervention. No significant differences were found. In regard to overall perceived impact (sentiment), more than half (54%; *n* = 57) of the respondents described that COVID-19 had affected their life negatively. In addition, 21% (*n* = 22) made statements reflecting both positive and negative impacts. Thus, 75% of the respondents described experiencing at least some negative impacts from the pandemic. For example, one respondent stated, “I worry every day about getting COVID-19. How would I take care of myself if I got it because I live alone and I would not want to jeopardize someone else’s heath to come and to take care of me. I’m trying very hard to stay positive and be as careful as I can for my health and for my mothers. I’m very nervous about the winter with the combination of COVID-19 and the flu.” A much smaller number (10%; *n* = 10) was coded as describing positive impacts only, and 15% (*n* = 16) were coded as neutral, indicating they reported no change or they made factual statements devoid of any discernible sentiment or affect (e.g., I am working more) (see Figure 1).

### 3.4. Frequency and Sentiment of Individual Themes

The eight core themes identified in the open-ended responses varied in the frequency with which they were mentioned as well as their associated sentiment. (See Figure 2). A deeper look into each theme provided greater insight into the variability in the impact of COVID-19 among women with SCI in this study. Each theme is discussed below, highlighting the percentage of women who mentioned each theme and the sentiments reflected in those responses (see Figure 3 for stacked view of sentiment by theme).

#### 3.4.1. Physical Health

Physical health was defined as statements focusing on one’s physical body, physical health care, or aspects of the woman’s health conditions (e.g., keywords included sick, wellness, illness, infection, health behaviors, medical appointments, physical therapy). Only 16% (*n* = 17) of the women responding noted physical health in their responses. Of these, half (52%; *n* = 9) reported that COVID-19 had a negative impact on their health. For example, one woman stated “I have a pressure sore that needed to be debrided but I was afraid to go to the wound care center. It got infected and started tunneling…”. A smaller percentage of women described physical health impacts that were coded as positive (17%; *n* = 3), mixed (12%; *n* = 2), or neutral (17%; *n* = 3).

#### 3.4.2. Mental Health

Mental health was defined broadly to capture statements in which participants expressed feelings or emotions in their response to the impact of the COVID-19 pandemic on their life experience. This included feeling overwhelmed, happy, sad, relaxed, scared, and afraid. It also included the experience of mental health conditions, including anxiety and depression. The impact of COVID-19 on mental health was the most frequently mentioned theme, noted by over 70% (*n* = 76) of the women with SCI in the study. Of these women, 61% (*n* = 47) expressed negative sentiments, suggesting an increase in feelings of anxiety, fear, and frustration. Negative sentiment codes for mental health often reflected feelings of sadness or fear related to not seeing loved ones, worry that they themselves or family members would get sick, or expressions of frustration that the community did not seem to be following health guidelines and were not thinking about the health risks that women with SCI may be experiencing. One woman stated, “Covid has affected my life in a negative way. Scared to be in public places, (grocery stores, etc.) in fear that I might contract the virus. Which then raises my daily anxiety”. Although 20% (*n* = 15) reported a mix of positive and negative mental health sentiments, a smaller percentage of women indicated only positive (8%; *n* = 6) or neutral (10%; *n* = 8) impacts of the pandemic on their mental health. For example, one woman stated, “Honestly, mostly positive ways, which feels terrible to say! My wife was furloughed for 6 weeks, so we basically got an extended honeymoon where we both couldn’t go anywhere but the house! And now that she’s back at work, she’s doing it from home, so I have my favorite person around all of the time. It’s SO helpful for my mental health!”.

#### 3.4.3. Social Health

Social health included statements noting an increase or decrease in social activities (e.g., daily contact, isolation, social support). Over half (55%, *n* = 58) of the respondents made reference to the pandemic’s impact on their social health. Of these, the majority of sentiments were coded as negative (59%, *n* = 34). These included negative sentiments about not being able to see friends and family. One woman stated, “It has dramatically impacted my ability to leave the house and to travel…it definitely has negatively impacted my social interactions”. Nonetheless, it is interesting to note that 22% (*n* = 13) reported positive changes in regard to social health during the pandemic. These included greater social connections on social media, including those made through zoom calls and face time. One woman stated, “The companionship, love, -of and from- my family, gives me much support and comfort during this precarious time of keeping COVID at bay”. Only 12% (*n* = 7) reported neutral social sentiments and 7% (*n* = 4) reported mixed sentiments.

#### 3.4.4. Activities of Daily Living (ADL/IADL)

This theme consisted of statements made regarding activities of daily living (ADLs; e.g., bathing, dressing) or instrumental activities of daily living (IADLs; e.g., cooking, shopping). Only 14% (*n* = 15) wrote about changes in this area. This was equally distributed across positive (*n* = 5), negative (*n* = 5), and mixed (*n* = 5) sentiments. Some positive comments involved receiving assistance. For example, one individual commented that she learned to order her groceries online and enjoyed this independence. Negative sentiments focused on difficulty obtaining help, for example “…it has been hard for… activities of daily living like hair/nails which may be luxury for some is essential to someone with limited hand function…” In addition, another woman wrote, “I am experiencing increased … difficulty obtaining essential items and services I need”.

#### 3.4.5. Exercise

The exercise theme referred to aspects of physical exercise, working out, physical fitness, and engaging in exercise inside or outside the home. Only 8% (*n* = 8) of respondents brought up aspects of exercise in their statements. Of these, only 25% (*n* = 2) were negative sentiments, while 75% (*n* = 6) were positive. For example, one woman stated, “… it allowed me to exercise more because I dont have to commute to school anymore…”. Another woman stated, “I remained active by going to the city trails and walking (rolling) and hour plus almost daily and am working twice a week with a personal trainer in his personal gym where everything is wiped down before, during and after my work out (including my chair!) I’m the strongest I’ve been in a very long time!”. There were no statements related to one’s ability to exercise that were coded as mixed or neutral.

#### 3.4.6. Work

The work theme referred to aspects of work such as working for pay (in or outside the home) or volunteer work. Thirty percent (*n* = 31) reported impacts on their work. Of these reports, some women reported they lost jobs and struggled to find new employment (negative = 45%; *n* = 14). One woman wrote, “Because of COVID I haven’t been able to work as much which affects how much I can do/spend”. However, several found the ability to work from home more accommodating than prior to COVID-19 (positive = 19%; *n* = 6). In addition, as expected, 32% (*n* = 11) made statements that were considered neutral, indicating there was no change in their work experience. One woman with an SCI stated, “…I already worked from home. So, it hasn’t affected my ability to work”. No women reported mixed sentiments regarding the impact of COVID-19 on work.

#### 3.4.7. Activities Outside of the Home

This theme referred to aspects of activities being done outside the vicinity of the home, such as going out, leaving the house, and travel. Thirty-nine percent (*n* = 41) of the respondents talked about changes COVID-19 had on their ability to participate in activities outside the home. Over half (59%; *n* = 24) of these were negative. Most of these were due to the pandemic-imposed restrictions and women with SCI’s inability to get out and do the things they used to do. For example, one woman wrote, “Due to covid, I and my family have had to limit our activities outside of the house to have lower exposure risks. Where I used to get out 3 to 4 times a week to run errands during the day I now only get out once a week on average”. Interesting to note, 32% (*n* = 13) stated that COVID-19 had not impacted their participation outside the home at all, often commenting that their activities outside of the home were already restricted prior to the pandemic. Positive sentiments, mentioned by only three (7%) respondents, included statements by individuals who decided to go out and be active regardless or those who reported that they enjoyed the “excuse” of not having to go out and participate in activities. Finally, only one respondent shared a mixed sentiment related to COVID-19’s impact on their activities. …I live in a very rural area of my state so we have not have high numbers of confirmed cases. That being said I still have a great deal of fear of contracting COVID. I did not go anywhere for 12 weeks. I just started going [out] the last two weeks which has been a mixed experience of good and bad. I am wearing a mask and using hand sanitizer and social distancing when I do get out.

#### 3.4.8. Activities at Home

This theme included activities being done inside or on the home premises, such as gardening, caring for pets, crafting, or reading. Only 8% (*n* = 8) of respondents included information about their participation in activities inside the home in their statements. Of these, the majority (63%; *n* = 5) were positive. Even though this is a very small percentage of the total number of respondents, it is important to note the positive impact individuals reported due to changes in response to COVID-19. For example, one woman wrote “I enjoy my computer, social media, movies, TV, reading, and cooking while at home”. Out of the remaining respondents, two were neutral and one reported a mixed sentiment regarding COVID-19’s impact activities in the home.

### 3.5. Relationship of Demographic and Disability Characteristics

We evaluated if the sample demographics (e.g., Injury Level) influenced sentiment ratings. Chi-square analyses were conducted on the categorical variables (Employment, Education, Race, Injury Level [paraplegia or tetraplegia]), and an ANOVA was conducted on the continuous variables (Age, Time Since Injury). These analyses were conducted across all sentiment ratings and then repeated by collapsing ratings into two groups (negative impact versus all other sentiment ratings). No significant differences were found between the demographic variables and the sentiment ratings.

## 4. Discussion

This study represents the first known examination of impacts of the pandemic on the lives of women with SCI residing in community settings within the United States. Using an online, qualitative approach, we asked a sample of women with traumatic SCI to respond to one open-ended question, “Please tell us how COVID-19 has affected you and your life”. Using Hand-coded Sentiment Analysis to identify the sentiment (positive, negative, mixed, or neutral) attributed to respondents’ statements, the majority of participants described negative impacts from the COVID-19 pandemic.

Openly sharing their stories and concerns, our respondents clearly perceived a wide range of impacts as demonstrated by their statements that resulted in eight major themes indicating areas of importance. Most respondents reported impacts on mental and social health, with the vast majority of them reporting negative impacts on mental as well as social health. They described increased feelings of loneliness, anxiety, and frustration, as well as reduced social interactions, a finding consistent with other research on the negative effect of the pandemic on the psychosocial health of people with SCI [19,23,25,30,36].

Many respondents identified impacts on Activities Outside of the Home. Most noted negative disruptions, which is not surprising given the implementation of social distancing and other disease mitigation measures during our data collection period. For example, several respondents cited missing physical exercise or physical therapy due to the temporary closing of facilities and the suspension of in-person services. Respondents also lamented the loss of ability to engage in activities outside the home, such as not being able to eat in restaurants, leave their homes, or travel. A minority, however, tended to focus on the positive impact of the pandemic and mentioned enjoying outside activities such as gardening or enjoying nature. Interestingly, nearly a third of the respondents who mentioned participation outside the home in their open-ended response indicated a neutral sentiment. This is likely because people with SCI often experience severely restricted activities outside of the home in pre-pandemic times due to functional limitations, inaccessible environments and transportation, lack of personal assistance and adaptive equipment, and other barriers to community participation.

A smaller number of respondents mentioned physical health and employment in their responses, but again, more identified negative impacts than positive. In fact, negative impacts on work were reported more than twice as often as positive impacts, and negative impacts on physical health were noted nearly three times as often as positive impacts.

While overall impacts were far more negative than positive, similar to results reported in a study on the effects of the pandemic on a predominantly male sample of people with SCI [37], some respondents in our study reported positive impacts. One in ten mentioned only positive impacts in their overall response, and an additional two in ten noted some positive impacts along with negative impacts in their overall response. Looking at the eight themes, positive impacts were more common than negative impacts on the low-frequency themes of exercise and activities inside the home. Notably, no responses related to participation inside the home were coded as negative. Positive impacts on activities inside the home included enjoying more time with family, re-engaging spiritually, organizing belongings, joining online groups, and painting. This finding is encouraging given that people with SCI experience limited and inadequate social engagement, loneliness, and participation in non-pandemic times [38,39,40]. In response to the pandemic onset, some organizations and providers began offering online, virtual services and supports, including online exercise class opportunities. Combined with the creativity and adaptability shown by this group of women, such provisions may have contributed to the positive responses pertaining to participation inside the home.

Although many reported serious negative impacts, at the time of this study, respondents appeared to have been spared other significant, if not life threatening, difficulties experienced by other people with disabilities. For example, while others reported food insecurities during the pandemic [41,42], none of our respondents identified food scarcity as a problem. Since our sample of women had the resources to participate in an online study, perhaps they also had the financial advantages to secure food. Additionally, only two respondents mentioned a problem accessing essential or medical supplies, and none referenced concerns about medical rationing—both difficulties that have been linked with a negative perceived impact of the pandemic on the health of people with SCI [20]. Notably, none of the respondents disclosed experiences with domestic violence, which is highly prevalent among people with disabilities [43] and has been referred to as a shadow epidemic during the COVID-19 crisis [44]. Moreover, although some respondents mentioned a family member or friend having COVID-19, none disclosed that they themselves had been infected with the virus. This may, in part, be a function of the timing of our data collection, which was relatively early in the pandemic. That said, other researchers have documented COVID-19 infection in approximately half of their samples of people with SCI [17,18]. Therefore, our findings may not adequately convey the full impact of the pandemic on people in this population that might have been captured over time, as the enormity of the pandemic was fully realized.

Nevertheless, our study has multiple strengths. For example, the focus is novel in that it offers a rare examination of the pandemic’s impacts on women with SCI, a population that is poorly represented in the SCI literature and seemingly missing entirely from general COVID-19 literature. Additionally, our study digresses from most of the existing research on the pandemic in the context of SCI that prioritizes disease management and prognosis, as noted by Morgan and colleagues [23]. Another strength is that the respondents used an open-ended survey question, allowing the respondents to share positive as well as negative impacts. Moreover, the data were collected three to eight months after the World Health Organization officially declared COVID-19 a global pandemic [45]. As such, the data reflect experiences in a unique period of time prior to the release of the COVID-19 vaccines and during a period of physical distancing, lockdowns, reduced access to health and disability services, and other restrictions.

Our study has several limitations. First, while some respondents provided lengthy responses, others were fairly brief and may have only mentioned the most salient impacts. Our methodology did not allow us to probe respondents’ statements to gain a more in-depth understanding of the impacts of the pandemic. Although our sample size is generous for including women with SCI, the small sample size may have limited our ability to detect significant differences between our study demographics and identified themes. Additionally, since we aimed to gather rich data on the perceptions of the impacts of the pandemic on women with SCI, the results are meaningful but not generalizable to other women or to people of other genders with SCI. Moreover, although our sample was diverse in terms of age, disability characteristics, and geographic representation, it was not fully representative in terms of level of education, race, and resources. Given the dynamic changes occurring over time (development of vaccines, reopening of schools and businesses, impact on healthcare providers and services), our study would be difficult to replicate. Furthermore, our predominantly non-Hispanic, White, relatively well-educated sample of women who were required to have access to a computer with high-speed internet for participation in the parent study is not representative of the population of women with SCI in the United States, who, as women with physical disabilities, are more likely to live below the poverty level, more likely to have only a high school education or less, and less likely to be employed compared to the general population of women [46].

There is a critical need for future research designed to systematically explore experiences in later stages of the pandemic and its prolonged burdens on the lives of women in this population. We call upon researchers to partner with people in the SCI community to examine and delve more deeply into the issues identified by our respondents and expand this work by involving a more diverse sample of women with SCI via the inclusion of more women who are members of racial and ethnic minorities and other marginalized communities. Additionally, research is needed that is inclusive of women with SCI who experience severe cognitive or sensory limitations or who live in nursing homes and other institutions precluding their eligibility for the current study.

## 5. Conclusions

Overall, the COVID-19 pandemic has had a significant impact on women with SCI. Our findings bring forward the complex and nuanced impact experienced by women with SCI, with both positive, negative, mixed, and neutral impacts. Although, the largest impact on women with SCI has been identified as negative, this study brings forward other reactions and impacts in this population that should be explored further. The eight themes identified in this study indicate areas of importance in this population, with mental and social health concerns being most prominent. The COVID-19 issues experienced by people with disabilities during the pandemic may persist as the virus becomes endemic post-pandemic. Those issues include elevated risks and fears related to exposure to the virus; need for equitable access to testing and vaccines; early intervention if infected, as well as a greater probability of experiencing severe illness, hospitalization, longer recovery process, psychosocial distress, and mortality (16–24). This new phase of the COVID-19 virus will continue to affect our communities, and there will be a need to address it in predictable ways, such as the measures taken to contain seasonal influenzas and their historically disproportionate consequences for people living with SCI and other chronic, disabling health conditions. We strongly urge providers, policy makers, health and disability researchers, disability advocates, and other influential entities to help ensure that, during future public health crises, women with SCI have access to disability-sensitive and affordable support, resources, and interventions to prevent the deterioration of their preexisting compromised health issues. Such mobilization of resources is especially important for women with SCI that lack adequate assistive devices and personal assistance, live in poverty and/or live in rural areas, and thus lack access to appropriate sources of assistance, technology, and opportunities.

## Figures and Tables

**Figure 1 ijerph-20-06387-f001:**
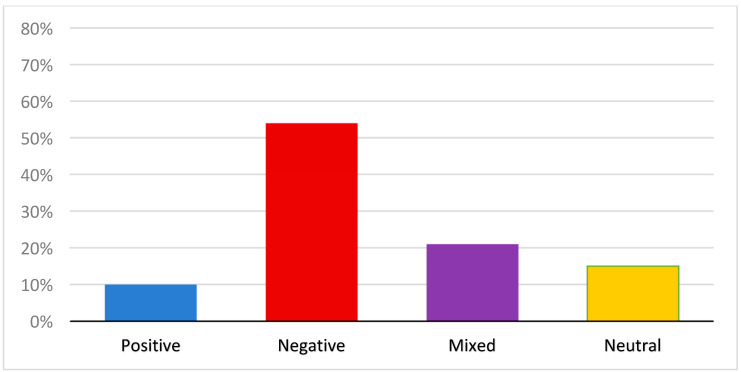
Percentage of overall impact statements by sentiment.

**Figure 2 ijerph-20-06387-f002:**
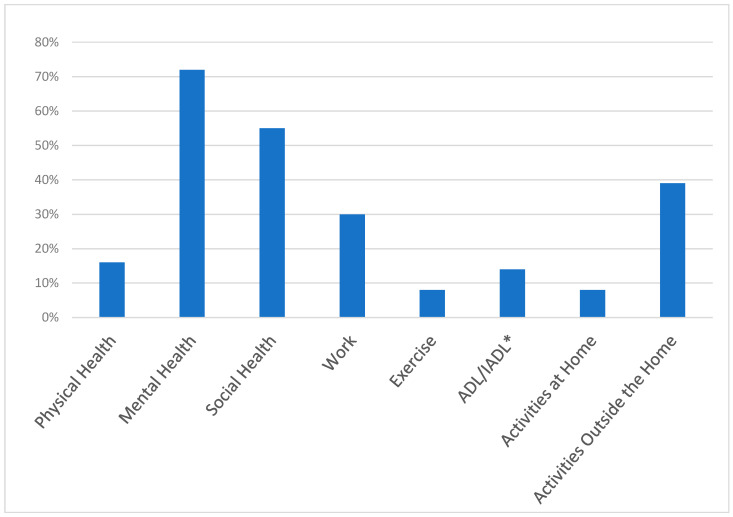
Percentage of responses by theme. * ADL, activities of daily living; IADL, instrumental activities of daily living.

**Figure 3 ijerph-20-06387-f003:**
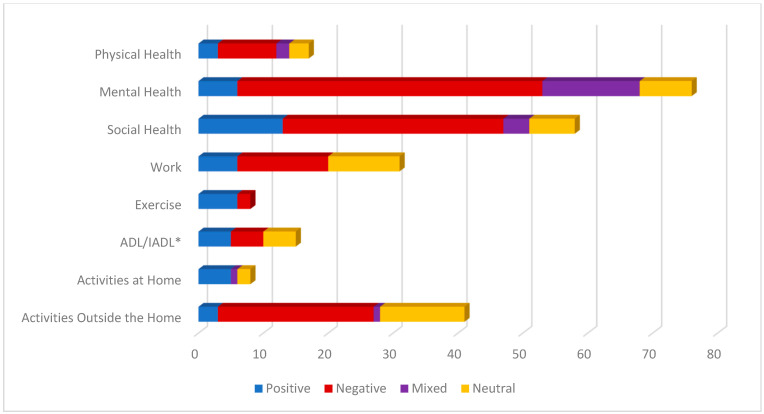
Number of respondents expressing sentiments by theme (*n* = 105). * ADL, activities of daily living; IADL, instrumental activities of daily living.

**Table 1 ijerph-20-06387-t001:** Sample characteristics (*n* = 105).

Variable	Mean (SD) or *n* (%)
Age (y)	43.20 ± 13.09
Race	
White	85 (81.0)
Black	9 (8.6)
Native American	1 (1.0)
Asian	2 (1.9)
Multiracial	7 (6.7)
Missing	1 (1.0)
Ethnicity	
Not Hispanic	96 (91.4)
Hispanic	9 (8.6)
Level of Education	
Less than high school	2 (1.9)
High school grad, GED	10 (9.5)
College, less than 4 years	29 (27.6)
Associate or bachelor’s degree	40 (38.1)
Master’s/doctoral degree	24 (22.9)
Employment Status	
Full-time	17 (16.2)
Part-time	23 (21.9)
Unemployed	65 (61.9)
Relationship Status	
Married	30 (28.6)
Unmarried couple	18 (17.1)
Single, never married	34 (32.4)
Divorced	18 (17.1)
Separated	1 (1.0)
Widowed	4 (3.8)
Household Income	
<$15,000	19 (18.1)
$15,000–$24,000	15 (14.3)
$25,000–$49,000	20 (19.0)
$50,000–$74,000	16 (15.2)
$75,000–$99,999	11 (10.5)
Over $100,000	18 (17.1)
Don’t know	6 (5.7)
Community Environment	
City or large town	43 (41.0)
Suburb or just outside a city or large town	29 (27.6)
Small town	25 (23.8)
The country or a long way from town	8 (7.6)
Level of Injury	
Paraplegia	56 (53.3)
Tetraplegia	49 (46.7)
Time Since Injury	17.47 ± 13.09
Primary Locomotion	
Power wheelchair	42 (40.0)
Manual wheelchair	57 (54.3)
Ambulatory	6 (5.7)
Personal Assistance Needed	
No Help Needed	34 (32.4)
Help needed with ADL OR IADL	28 (26.7)
Help needed with both ADL and IADL	43 (41.0)

Note. ADL = Activities of Daily Living; IADL = Instrumental Activities of Daily Living; GED = General Educational Diploma.

## Data Availability

Data can be requested by contacting the corresponding author. Data are unavailable publicly due to privacy restrictions.

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
