# Peer review of "Psychosocial Impacts of the COVID-19 Pandemic on Women with Spinal Cord Injury"

_ijerph, 2023, doi:10.3390/ijerph20146387_

Round 1

Reviewer 1 Report

The study focuses on exploring the perceived positive and negative impacts of the COVID-19 pandemic on women with spinal cord injury (SCI).  The article highlights the challenges faced by women with SCI, including increased stress, depression, and loneliness due to social isolation, and substantial barriers to accessing medical supplies, food security, healthcare services, and personal assistance services. The study is significant because it is the first known examination of the impact of the pandemic on women with SCI residing in community settings. The study used an online, qualitative approach, which allowed women to express their views on how COVID-19 has affected them and their lives. The authors reported eight major themes indicating areas of importance, including impacts on mental and social health, activities outside the home, physical health, and employment. While most respondents reported negative impacts, some respondents reported positive impacts, such as enjoying more time with family, re-engaging spiritually, organizing belongings, joining online groups, and painting.

Although the study has many strengths, including its focus on a population that is often overlooked in SCI literature and general COVID-19 literature, it has some limitations. For example, the results are meaningful but should be compared with previous studies on people including other genders with SCI. Additionally, the sample was not fully representative in terms of level of education, race, and resources.

Provide more details about the sample: While the article mentions that the sample consisted of paraplegic and tetraplegic, no differences are reported. Sentiment analysis is not always a reliable method for understanding the complexities of human experiences; please comment on this. 

The article provides a thorough analysis of the impacts of the pandemic on women with SCI, but it does not discuss the implications of these findings for policy and practice. The authors could consider discussing policy and practice recommendations based on their findings.

Reviewer 2 Report

This manuscript is well written. Nevertheless, I have a few points for consideration by the authors and editor.

 1.      I would try to include the 8 main themes that participants brought forward in the abstract.

2.    2.   I have to say that I consider the manuscript a little long and wordy.  I would try to reduce the word count.  Particularly in the introduction. 

3.      3. The study is a little “taking advantage of an opportunity”. Covid was not planned but when it took place the authors were able to fit an extra question in to an existing study. The authors point out that their study took place at the beginning of the pandemic which this reviewer first thought was unfortunate because people had only been in quarantine a few months and the real effects on mental health had not been felt yet.  There was no vaccine and no date for its development.  There was no discussion about being vaccinated or not and eg travel requirements related to vaccination.  So a large number of concerns and stress factors that came later in the pandemic were not yet present.  Nevertheless, the authors to their own defense point out that the rules in the period were clear: stay home. Schools and businesses were closed. So this made the study background more clear.

4.     4.  What is not clear is the recruitment of participants.  How many persons were recruited but did not take part?  The authors rightly point out as can be expected that participants were better than average educated and financially well off. (they had high speed internet at home). They were also volunteers, which might mean that they want to give positive answers or want to participate to tell someone about their complaints.  So, I think there could be a little more discussion about the limitations of the participant group.

5.     5.  There is nothing about the difference between those who provided written answers to the question and those who were interviewed by phone.  An interviewer can try to get more out of the respondent?

6.      6. Finally I am always wondering if it was not possible to give us some insight into the differences in response per income level, employed or unemployed (more than 50% were unemployed but are these also the low income group?  Are unemployed less mobile?  Single, married or otherwise?  Education level?  Time since injury (recent or long ago?). Are the subgroups different in response. Do older people answer differently? ( I would give the age range in the table and not only in the text and maybe even a frequency table of age groups).

7. I do think that the authors need to point out someplace that because the covid pandemic is now in a completely different phase, that it is actually impossible to reproduce this all. And even make a suggestion as to what we can do now to confirm or refute the findings.  We need to prepare for a new pandemic in the future but do we know enough now based on this study to make a clear plan.  Maybe a few sentences of reflection.

Reviewer 3 Report

1. The format is not aligned with the journal’s official format, please correct it.

2. 2.1 sample and 3.1 participants can be integrated.

3. Data processing procedure needs to be stated clearer.

4. The overall construction of the paper is chaotic, please re-organize. Especially, 2. Method section.

5. It’s hard to catch the whole picture of 2. Method section and 3. Results. In Method section, the author should clearly state participants, tools, variables, research hypothesis or research questions, and data processing. Results section extended many sub-sections that did not mention in 2. Method section. It’s hard to catch what the author wants to research.

6. Conclusions should be stated clearer for highlighting the contributions. In addition, please mention the limitations of the current and suggestions for future research.

7. The manuscript should be proof read, otherwise it is difficult to understand and read. Please re-check the grammar and spelling.

Round 2

Reviewer 1 Report

Thank you for author comments Manuscript technically sound. 

Author Response

Thank you for your comments